# Morphological and Molecular Characterization of Quinoa Genotypes

**Ehab H. EL-Harty, Abdelhalim Ghazy, Talal K. Alateeq** **, Sulieman A. Al-Faifi, Muhammad Altaf. Khan, Muhammed Afzal** **, Salem S. Alghamdi and Hussein M. Migdadi ***

Plant Production Department, College of Food and Agricultural Sciences, King Saud University, P.O. Box 2460, Riyadh 11451, Saudi Arabia; eelharty@KSU.EDU.SA (E.H.E.-H.); aghazy@ksu.edu.sa (A.G.); talateeq@ksu.edu.sa (T.K.A.); salfaifi@ksu.edu.sa (S.A.A.-F.); altaf_sbs@yahoo.com (M.A.K.); mmushtaq@ksu.edu.sa (M.A.); salem@ksu.edu.sa (S.S.A.)
* Correspondence: hmigdadi@ksu.edu.sa

**Abstract:** Quinoa cultivation has expanded from South America to many countries because of its wide adaptability and nutritional value. We evaluated 32 introduced quinoa genotypes using 17 qualitative and 11 quantitative traits under Saudi Arabia conditions during the 2018–2019 season. The quinoa genotypes showed considerable variation during the vegetative and maturity stages. Plant height values varied between 60 and 18 cm, and maturity ranged from 98 to 177 days. Leaf shapes were rhomboidal or triangular, with dentate or serrate margins. Green was the standard color for leaves and panicles at the flowering stage. The leaf granule colors were white, purple, and white-red. At 150 units of the Euclidean distance, the genotypes aggregated into four major groups based on their morphological traits. Twenty-one sequence-related amplified polymorphism (SRAP) primer pair combinations generated 75 amplified fragments (alleles), with a mean of 3.57 alleles per primer pair combination. Unweighted Pair-Group Method with Arithmetic means (UPGMA) clustering analysis showed that the quinoa genotypes were grouped based on origin or according to genetic background. Genotypes from South America presented higher mean values for the average number of alleles, Shannon index (0.411), gene diversity (0.271), and polymorphic percentage (83.95). Analysis of molecular variance (AMOVA) showed that most of the genetic variation was because of differences within populations (86%). The wide variability of the genotypes studied herein is of great importance for quinoa breeders.

**Keywords:** quinoa; SRAP; morphology; molecular markers; biplot; cluster analysis; AMOVA

## 1. Introduction

Quinoa (*Chenopodium quinoa* Willd.) is an annual crop first cultivated by the Incas over 7000 years ago [1]. Its cultivation area has increased notably in the origin area (Andean region) and has spread to North America, Europe, Africa, and Asia because of its adaption to various conditions, including drought, salinity, frost, and marginal soils [2]. Bazile et al. and Tapia classified quinoa into five ecotypes based on their geographical adaptations and distribution. Valley ecotypes are grown at 2000 to 3500 m, and those grown at high altitudes above 3500 m, around Titicaca Lake, belong to Altiplano ecotypes. Ecotypes grown in the salt flats of Bolivia and Chile are called Salares. The fourth ecotypes, grown in low-altitude areas of southern and central Chile, are called sea-level ecotypes, while the subtropical ecotypes are grown at low-altitudes [1,3]. This adaptability suggests that quinoa will become an alternative crop in marginal environments and regions affected by climate change [2]. It can grow in the arid environments of the semi-desert regions of South America and other similar regions worldwide. It is considered a drought-tolerant crop, capable of developing and producing seeds with less than 200 mm of annual rainfall [4,5]. Quinoa belongs to halophyte group of crops; Schmockel et al. identified 15 genes that could contribute to the differences in salinity tolerance of tested quinoa accessions [6].

Its grains have high nutritional value, with higher lysine (5.1–6.4%) and methionine (0.4–1.0%) contents, thus having a better balance of essential amino acids than cereals and legumes [7]. The introduction and evaluation of breeding materials are the first steps in any breeding program.

The success of emerging breeding programs will rely on collecting and assessing these genotypes [8]. Quinoa possesses a vast morphological variation; agronomical studies are essential to knowing these exotic genotypes' agronomic potential and their use in plant breeding. Additionally, identifying each quinoa variety by morphological characterization will help in better genotype selection for farmers and marketers. Zhang et al. reported that breeding to improve quinoa was restricted because of a lack of genetics and genomic information about the crop. They suggested that the knowledge of the plant's genomic variation, population structure, and genetic diversity is necessary [9]. Analyzing genetic diversity will help to assess the conservation status and strategies used for these valuable resources. It enables us to find additional allelic variation sources to increase crop productivity and geographic range [10]. Genetic markers are crucial for germplasm conservation and core collection development. Several studies have classified quinoa genotypes using molecular marker technologies [8,9,11,12]. Sequence-related amplified polymorphism (SRAP) markers were developed by Li and Quiros [13] and are used to amplify the coding regions of DNA, with primers targeting open reading frames. Sequence-Related Amplified Polymorphism (SRAP) markers are used in research addressing plant systematics and biogeography hypotheses and are applying in many plant biological studies. Robarts and Wolfe [14] summarized the results of 171 publications using SRAP markers. They highlighted potential of SRAP markers to enhance the current suite of molecular tools in various fields by providing an easy-to-use, variable marker with inherent biological significance [14]. This is the first study reporting the use of SRAP for assessing quinoa diversity, to the best of our knowledge. This study was aimed at characterizing 32 introduced genotypes of quinoa grown under the semi-arid climatic conditions of Saudi Arabia at the morphological level and to assess genetic diversity of these genotypes at the molecular level using SRAP markers.

## 2. Materials and Methods

### 2.1. Plant Materials

Thirty-two quinoa genotypes were used in this investigation, comprising landraces, improved lines, and cultivars introduced from seven countries and six states of the USA. The genotypes' seeds were provided from the U.S. National Plant Germplasm System and Giza 1 genotype from the Agriculture Research Center. The names, origins, and sources of the quinoa genotypes are presented in Table 1.

### 2.2. Field Experiment

Genotypes were planted on 1 December 2018 at Dirab Experiments and Agricultural Research Station, Riyadh (24°43′34″ N, 46°37′15″ E). The experimental soil was sandy clay loam (pH = 8.15; electrical conductivity = 2.1 dS m$^{-1}$). Each genotype was planted in a plot comprising rows 3 m long, with an inter-row distance of 50 cm and an intra-row plant spacing of 20 cm. After seedling's emergence, plants were thinned out, leaving one plant per hole. Plots were kept free of weeds through hand hoeing twice during the vegetative period. Calcium superphosphate ($CaH_6O_9P_2$) was applied during soil preparation at a rate of 71.4 kg $P_2O_5$/ha. Nitrogen as ammonium sulfate (60 kg/ha) was applied in two equal doses: the first with sowing and the second at four weeks after planting. After full emergence, plants were watered once per week to avoid drought stress, and quinoa plants were covered after flowering, using plastic nets to protect plants from birds. Seventeen qualitative and 11 quantitative traits were scored based on the descriptions provided by [15]. The qualitative traits included growth habit (GH), stem shape (SS), stem color at maturing stage (SC), pigmented axis (PA), striae color (StC), the position of branches (PBr), leaf shape (LS), leaf margin (LM), leaf color (LC), leaf granule color (LGC), panicle

color at flowering (PCF), panicle color at maturity (PCM), panicle shape (PS), panicle density (PD), dehiscence degree (DD), perigonium appearance (PG), and perigonium color (PC). Quantitative measurements included plant height (PH), stem diameter (SD), number of branches/plant (NoB), panicle length (PL), panicle width (PW), petiole length (PeL), leaf length (LL), leaf width (LW), leaf area (LA), the number of teeth/leaf (NoT), and the number of days from sowing to 95% maturity (MD).

**Table 1.** The genotype name, ID number, origin, and status of each of the 32 quinoa genotypes used in the study.

| No. | Accessions | ID.Number | Origin | Status |
|---|---|---|---|---|
| 1 | Giza1 | — | Egypt | Cultivar |
| 2 | LP-128 | PI 587173 | Argentina | Cultivar |
| 3 | Q-Silvestre | PI 510547 | Argentina | Cultivar |
| 4 | Bianra-de-Juny | PI 665272 | Australia | IG |
| 5 | Apelawa | Ames 13747 | Bolivia | IG |
| 6 | CQ-125 | PI 614925 | Bolivia | IG |
| 7 | Pasan-Ralle | PI 470932 | Bolivia | IG |
| 8 | Sayana | PI 614922 | Bolivia | Cultivar |
| 9 | Line-0692 | PI 665275 | Bolivia | IG |
| 10 | Q-Sajama-Jusi | PI 510545 | Chile | Cultivar |
| 11 | Pichaman | PI 634919 | Chile | Landrace |
| 12 | QQ-87 | PI 614884 | Chile | Landrace |
| 13 | UDEC-3 | PI 634925 | Chile | Landrace |
| 14 | DE-1 | PI 674266 | Ecuador | Cultivar |
| 15 | Grande | PI 510540 | Peru | Cultivar |
| 16 | Q-de-Quiaca | PI 510532 | Peru | IG |
| 17 | QQ-065 | PI 614880 | Peru | Cultivar |
| 18 | QQ-61 | PI 614888 | Peru | Landrace |
| 19 | Quinua | PI 510551 | Peru | Cultivar |
| 20 | Q-Amarillo | PI 510543 | Peru | Cultivar |
| 21 | Q-Blanca | PI 510548 | Peru | Cultivar |
| 22 | Col-#6197 | PI 665283 | USA, Colorado | IG |
| 23 | Colorado-407D | PI 596293 | USA, Colorado | Cultivar |
| 24 | 537-BK60-B | PI 677096 | USA, Maryland | Cultivar |
| 25 | 3P | Ames 13741 | USA, New Mexico | IG |
| 26 | Copacabana | Ames 13748 | USA, New Mexico | IG |
| 27 | Kaslaea | Ames 13745 | USA, New Mexico | IG |
| 28 | 37TES | Ames 13723 | USA, New Mexico | IG |
| 29 | 79R | Ames 13720 | USA, New Mexico | IG |
| 30 | NSSL-91567 | PI 677099 | USA, New York | Cultivar |
| 31 | NSSL-86649 | PI 677097 | USA, South | Cultivar |
| 32 | Japanese-strain | PI 677100 | USA, Washington | Cultivar |

Improved genotype (IG).

### 2.3. Molecular Characterization-Based SRAP Markers

Genomic DNA was extracted from bulked leaves of three individual plants per genotype using the CTAB extraction protocol described by [16]. SRAP-PCR amplification was performed according to the method proposed in [17]. Ninety-nine SRAP primer combinations were tested in a panel of eight DNA samples, while 21 SRAP primer combinations were selected based on polymorphism information and the number of generated alleles. In contrast, the primer combination generated less than three alleles or showed monomorphic patterns across the eight samples ruled out from the analysis (Table S1). The PCR assay was performed in 20 μL containing 1X GoTaq Green Master Mix (Promega Corporation, Madison, WI, USA), using 0.5 μM for each forward and reverse primer, 50 ng template DNA, and nuclease-free water up to 20 μL. Amplification of DNA was carried out on a TC-5000 thermal cycler (Bibby Scientific, UK). Amplification conditions were: first denaturation at 94 °C for 2 min, followed by five cycles of denaturation at 94 °C for 1 min, annealing at 35 °C for 30 s, and elongation at 72 °C for 45 s. A further 35 cycles of denaturation at 94 °C

for 30 s, annealing at 55 °C for 30 s, and elongation at 72 °C for 45 s were performed. A final extension step at 72 °C for 7 min was also performed. The amplification fragments were separated on 3% agarose gel with a constant voltage of 50 V following ethidium bromide staining (10 µg/mL). The sizes of DNA fragments were estimated using the 100 bp DNA Ladder size marker (Promega).

*2.4. Data Analysis*

Agglomerative hierarchical clustering (AHC) of genotypes was based on standardized Euclidean distances using qualitative traits and quantitative data after transformation to scale according to mean and standard deviation values using XLSTAT software [18]. Plant heights were scaled as short, medium, and long according to their mean and standard deviation values (1 = short, 2 = medium, and 3= long shoots).

Principal component analysis (PCA) was used to detect the morphological characteristics that explained the variation among the genotypes, and the percentage contribution of different morphological characteristics towards genetic diversity was calculated according to [19].

The gel-based SRAP fragment was checked and scored. All matrices were combined to form one binary matrix for further analyses, and the polymorphism information content (PIC) was calculated for each primer to estimate its allelic variation according to the formula described by [20]. Data generated from SRAP analysis were analyzed using Jaccard similarity coefficients [21], and the associations between the genetic dissimilarities were tested to create phylogenetic trees based on the unweighted pair group method. The relationships between the Euclidean distance matrix based on morphological traits and the genetic distance matrices obtained with SRAP markers were analyzed according to [22] using PAST software.

Allelic data obtained from fragment analysis were scored according to band presence/absence. STRUCTURE 2.3.4 software was used to detect the number of subpopulations explaining population structure [23]. The STRUCTURE analysis was run with parameters of a burn-in period of 50,000 and 50,000 MCMC replications, and a hoc statistic introduced by [24] was used to determine the correct estimated number of clusters with STRUCTURE harvester software (Earl and VonHoldt, 2012). The STRUCTURE harvester software was used to find the correct number (K) of subpopulations. K was tested from 1 to 12 with ten iterations for each group. ΔK was used to determine the correct cluster number. If the ΔK value is high, the probability of population cluster number is the most correct. Genotypes were assigned to a cluster if the probability of membership > 70%; if membership was < 70%, genotypes were assigned to the mixed cluster (admixture). The total number of alleles, genetic diversity (He), Shannon index for each population, and a number of private alleles per population were calculated. The genetic differentiation between populations was determined using phiPT a measure that allows intraindividual variation to be suppressed (heterozygosity). Analysis of molecular variance (AMOVA) among populations was performed using GenAlex 6.5 software [25].

## 3. Results and Discussion

The results of morphological traits showed tremendous variation among genotypes. These variations are essential for developing new cultivars with distinct morphologic and agronomic traits. Table 2 presents the distribution frequency of the quinoa genotypes' morphological characteristics, while the full description of each genotype is presented in Table S2. The growth habit (GH) divided quinoa genotypes into two dominant groups—the first group of genotypes branched from the base to two-thirds of the main stem (69% of genotypes), while the second group of genotypes branched to the panicle (31%); all branches (PBr) end with the panicle, but the panicle of the main stem was the largest. Branches were oblique (56%) or curved (44%) on the stem, as shown in Figure 1. Stem shapes (SS) were cylindrical (34%) or angular (66%). Although ten stem colors (SC) and three different stem striae colors (StC) were listed in the Bioversity International descriptor

for quinoa, only six SCs and three StCs were observed in this study. Green was the most common for both stem color (31% of genotypes) and striae color (66%). The leaves were green in 72% of genotypes; however, leaf granules (LGC), if present, were white, white-red, and purple in 63%, 6%, and 16% of genotypes, respectively (Figure 1). The leaf shapes (LS) were triangular (59%) or rhomboidal (41%) with a serrated margin (56%). The panicle shape (PS) descriptor classified genotypes into glomerulate (16%), intermediate (38%), and amarantiform (47%). Most genotypes were lax or intermediate for panicle density (PD) (50% and 44%, respectively).

**Table 2.** Categories and absolute and relative frequency distributions of discrete variables of morphological characters of tested quinoa genotypes.

| Trait | Category | Frequency Absolute | Frequency Relative (%) | Trait | Category | Frequency Absolute | Frequency Relative (%) |
|---|---|---|---|---|---|---|---|
| GH | Branched to 2/3 the main stem | 22 | 69 | PBr | Oblique | 18 | 56 |
| | Branched to the main panicle | 10 | 31 | | Slightly curved | 14 | 44 |
| SC | Green | 10 | 31 | PCM | Orange | 15 | 47 |
| | Yellow | 8 | 25 | | Yellow | 9 | 28 |
| | Red | 8 | 25 | | Red | 4 | 12.5 |
| | Other colors | 6 | 19 | | Other colors | 4 | 12.5 |
| StC | Green | 21 | 66 | PS | Glomerulate | 5 | 16 |
| | Red | 3 | 9 | | Intermediate | 12 | 38 |
| | Purple | 8 | 25 | | Amarantiform | 15 | 47 |
| LC | Green | 23 | 72 | PD | Lax | 16 | 50 |
| | Green-red | 8 | 25 | | Intermediate | 14 | 44 |
| | Red | 1 | 3 | | Compact | 2 | 6 |
| LM | Entire | 1 | 3 | DD | Light | 7 | 22 |
| | Dentate | 13 | 41 | | Regular | 19 | 59 |
| | Serrate | 18 | 56 | | Strong | 6 | 19 |
| SS | Cylindrical | 11 | 34 | PG | Semi-opened | 18 | 56 |
| | Angular | 21 | 66 | | Closed | 14 | 44 |
| LS | Rhomboidal | 13 | 41 | Pa | Absent | 19 | 59 |
| | Triangular | 19 | 59 | | Present | 13 | 41 |
| PCF | Green | 24 | 75 | LGC | Absent | 5 | 16 |
| | Purple | 2 | 6 | | White | 20 | 63 |
| | Red | 4 | 13 | | White-red | 2 | 6 |
| | Mixture | 2 | 6 | | Purple | 3 | 16 |
| PC | Cream | 11 | 34 | | | | |
| | Yellow | 3 | 9 | | | | |
| | Red | 6 | 19 | | | | |
| | Orange | 4 | 13 | | | | |
| | Other five colors | 8 | 25 | | | | |

Growth habit (GH), stem shape (SS), stem color at maturing stage (SC), pigmented axis (PA), striae color (StC), the position of branches (PBr), leaf shape (LS), leaf margin (LM), leaf color (LC), leaf granules color (LGC), panicle color at flowering (PCF), panicle color at mature (PCM), panicle shape (PS), panicle density (PD), dehiscence degree (DD), perigonium appearance (PG), and perigonium color (PC).

Quinoa genotypes were classified into four categories based on the panicle color at flowering (PCF), with a green present in 75% of genotypes; however, the green color could be changed at the maturing stage (PCM). In contrast, at the maturing stage, the panicle had six colors. Perigonium was present on the panicle and cover seeds during physiological maturity. The perigonium (PG) could be semi-opened (Figure 1) or closed (covering the grain in 44% of genotypes) with one of 15 colors, according to Bioversity International, although this study defined only nine colors for the perigonium (PC). Dehiscence degree (DD) is a measure of grain persistence in the plant at physiological maturity, and most genotypes (59%) had intermediate persistence of seeds. Bhargava and Ohri found various stem colors, branching types, seed colors, and panicle colors [26]. This diversity is reflected at the molec-

ular level and is used by plant breeders worldwide to develop improved genotypes—broad genetic variations among Ecuadorian landraces detected by morphological evaluations [3]. The mean for continuous variables under study showed wide variation between quinoa genotypes (Table 3), while the individual genotypes' mean performances are presented in Table S3. Plant height (PH) ranged from 60 cm for the Japanese strain to 180 cm for Line 0692. Tan and Temel estimated plant height at approximately 78–116 cm in Turkey [27]. The highest number of branches/plant (24) was produced by genotype Q. de Quiaca, while 50% of the genotypes were in the category with the highest number of branches, categorized into 17–23 branches/plant group; this attributed to the low plant density.

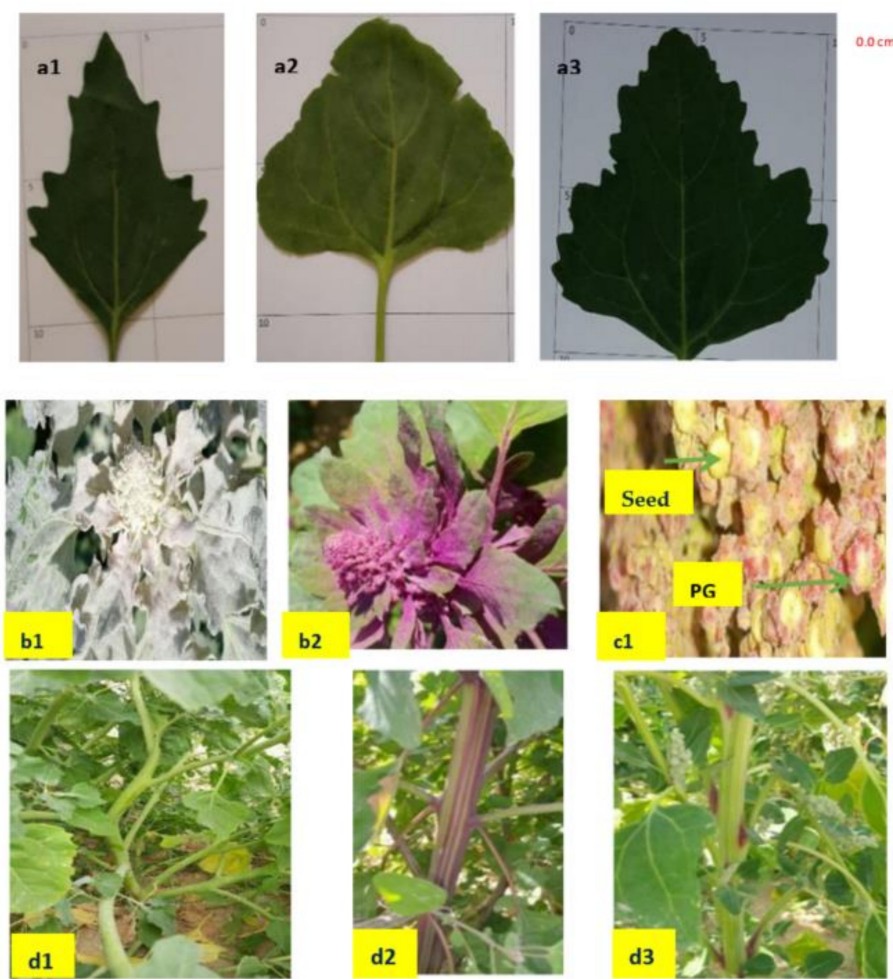

**Figure 1. a1**: Rhomboidal leaf shape with dentate margin. **a2**: Triangular leaf shape. with entire margin, **a3**: Triangular leaf shape with a high number of teeth on a serrate margin. **b1**: White leaf granules. **b2**: Purple leaf granules. **c1**: Semi-open perigonium. **d1**: Oblique branch position. **d2**: Stem striae. **d3**: Pigmented axis.

This study tested five quantitative quinoa leaf traits; petiole length (PeL), leaf length (LL), leaf width (LW), leaf area (LA), and the number of teeth/leaf (NoT). Leaf length (LL) values were between 6 and 12 cm, leaf width (LW) values ranged from 5 to 11 cm, and leaf area (LA) values ranged from 21 to 66 cm$^2$. Leaf margin (LM) groupings for quinoa were entire, dentate, or serrate, with teeth/leaf (NoT) values ranging from 6 to 21 teeth/leaf (Figure 1).

**Table 3.** Mean ± standard error (S.E.) and frequency distribution for each quantitative trait of the tested quinoa genotypes.

| Trait | Mean ± S.E. | Category | Frequency | |
|---|---|---|---|---|
| | | | Absolute | Relative (%) |
| PH (cm) | 107.4 ± 27.2 | 60–80 | 4 | 13 |
| | | 80–135 | 25 | 78 |
| | | 136–180 | 3 | 9 |
| NoB | 15.6 ± 4.4 | 8–11 | 5 | 16 |
| | | 12–16 | 11 | 34 |
| | | 17–23 | 16 | 50 |
| SD (mm) | 11.7 ± 4.3 | 8–12 | 8 | 25 |
| | | 13–20 | 21 | 66 |
| | | 21–27 | 3 | 9 |
| PeL (cm) | 7.2 ± 1.2 | 5–6 | 9 | 28 |
| | | 7–9 | 13 | 41 |
| | | 9–11 | 4 | 13 |
| LL (cm) | 8.9 ± 1.4 | 6–7 | 5 | 16 |
| | | 8–10 | 23 | 72 |
| | | 11–12 | 4 | 13 |
| LW (cm) | 7.8 ± 1.4 | 5–6 | 7 | 22 |
| | | 7–9 | 20 | 63 |
| | | 10–11 | 5 | 16 |
| LA (cm) | 40.4 ± 11.5 | 21–28 | 7 | 22 |
| | | 28–52 | 20 | 63 |
| | | 53–66 | 5 | 16 |
| NoT | 13.1 ± 3.8 | 6–9 | 7 | 22 |
| | | 10–16 | 19 | 59 |
| | | 17–21 | 6 | 19 |
| PL (cm) | 22.7 ± 4.4 | 17–19 | 5 | 16 |
| | | 20–26 | 23 | 72 |
| | | 27–32 | 4 | 13 |
| PW (cm) | 8.7 ± 2.3 | 5–6 | 4 | 13 |
| | | 7–12 | 26 | 81 |
| | | 13–15 | 2 | 6 |
| MD | 129.2 ± 21.7 | 98–107 | 6 | 19 |
| | | 108–151 | 19 | 59 |
| | | 152–177 | 7 | 22 |

Note: Plant height (PH), number of branches/plant (NoB), stem diameter (SD), petiole length (PeL), leaf length (LL), leaf width (LW), leaf area (LA), number of teeth/leaf (NoT), panicle length (PL), panicle width (PW), number of days from sowing to 95% maturity (MD).

The panicle length (PL) and width (PW) are essential traits for yield components, and these traits were in the ranges of 17–32 and 5–23 cm, respectively. The longest panicles were from two landraces (QQ 87 and QQ 61) from Argentina and Chile and two improved genotypes (Q. de Quiaca and Line 0692) from Peru and Bolivia, respectively. These latter two genotypes also had the widest panicles (32 cm). Bhargava et al. found significant genotypic and phenotypic correlation coefficients between plant height, branches/plant, and leaf area with seed yield/plant [28]. These show that selective breeding programs can improve quinoa productivity. Since the genotypes have different origins and need different day lengths and temperatures, their maturation periods also changed. Quinoa plants under Saudi Arabia conditions needed 98–177 days to mature after being sown. The earliest (matured after 98–105 days) genotypes were 37TES, Colorado 407D, and Giza 1 from New Mexico (USA), Colorado (USA), and Egypt. Quinoa genotypes grew to harvest maturity in 108–181 days in Denmark [29],109–163 days in India [28], and 107–158 days in Turkey [27]. The latest maturing genotypes (Q. de Quiaca and Line 0692) ranked first for plant height, leaf area, panicle length, and the number of branches/plant. Still, the early

maturity genotype has an advantage because of the increased risk of heat toward the end of the season.

These results agreed with previous results presented in [3], in which a broad genetic variation among Ecuadorian landraces was detected using morphological evaluations. To understand the contributions of the traits being studied to the variation among genotypes, a principal component analysis (PCA) was performed (Figure 2). The first and second components explained 37.9% and 27.3% of the total variation, respectively, with a total value of 65.2%. These axes show relevant discriminatory traits, including the perigonium color (PC), stem color at maturity stage (SC), striae color (StC), perigonium axis (PA), leaf area (LA), leaf length (LL), and stem shape (SS). The Q Amarillo, Apelawa, and 37 TES quinoa genotypes were grouped by perigonium color (PC). Bhargava et al. [28] estimated the variation among quinoa genotypes in the first PC from 39.5% with the most significant coefficients traits (plant height and stem diameter). The characteristics with a positive weight on PC2 were days to maturity, panicle length, and branches/plant. However, in [30] results, the first PC accounted for 74% of the variation among quinoa segregation generations. Morphological traits are used to assess the morphological diversity between quinoa genotypes grouped using Euclidean distance (Figure 3).

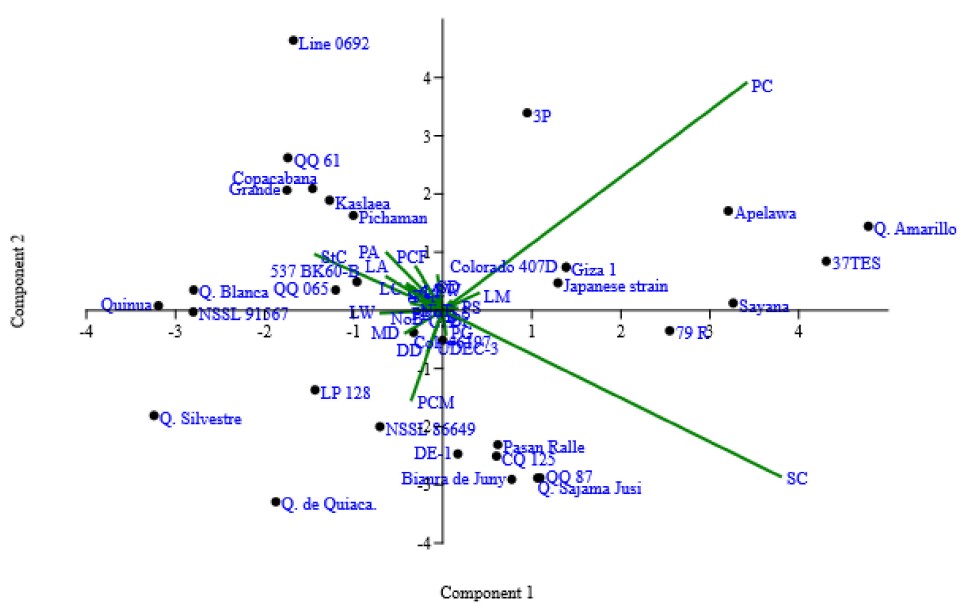

**Figure 2.** Two-dimensional ordination of the qualitative and rescaled quantitative data traits in quinoa genotypes. Component 1 represented 37.9%, and component 2 represented 27.3% of the total variance.

Cluster analysis using Unweighted Pair-Group Method with Arithmetic means (UP-GMA) and the standardized Euclidean distance coefficient was used to produce a dendrogram of quinoa genotypes, whereby the quinoa genotypes were categorized into four groups.

Five cultivars or improved genotypes (79 R, 37TES, Sayana, Q. Amarillo, and Apelawa) from the USA, Bolivia, and Peru were grouped in the first cluster, with 17.6% variance among them. The second cluster contained 20.0% of the total variance among 17 genotypes. In comparison, the third cluster grouped six genotypes from five countries and produced the lowest variation (9.8%). Four cultivars (NSSL 86649, Q. de Quiaca. Q. Silvestre, and 537 BK60-B) from the USA and Peru were collected in the fourth cluster, with a 20.0% variance among them [30].

Twenty-one SRAP primer pair combinations were used to assess the genetic distances among the tested quinoa genotypes. The SRAP primer combinations produced 75 amplified fragments (alleles), with a mean of 3.57 alleles per primer, while the total number of polymorphic fragments (bands) was 1105, with an average of 52.62 fragments for each primer combination (Table 4). Primer combinations ME19/EM19, ME20/EM22, and ME27/EM28 produced the most significant number of alleles (6), while primer combinations ME28/EM29, ME26/EM20, and ME097/EM19 produced the most significant number of bands (77, 74, and 74 fragments, respectively). The SRAP technique was developed by [13] and has emerged as a new and valuable marker technique for germplasm biodiversity. These results agreed with the previous publications on mung bean [31] and lupin [32].

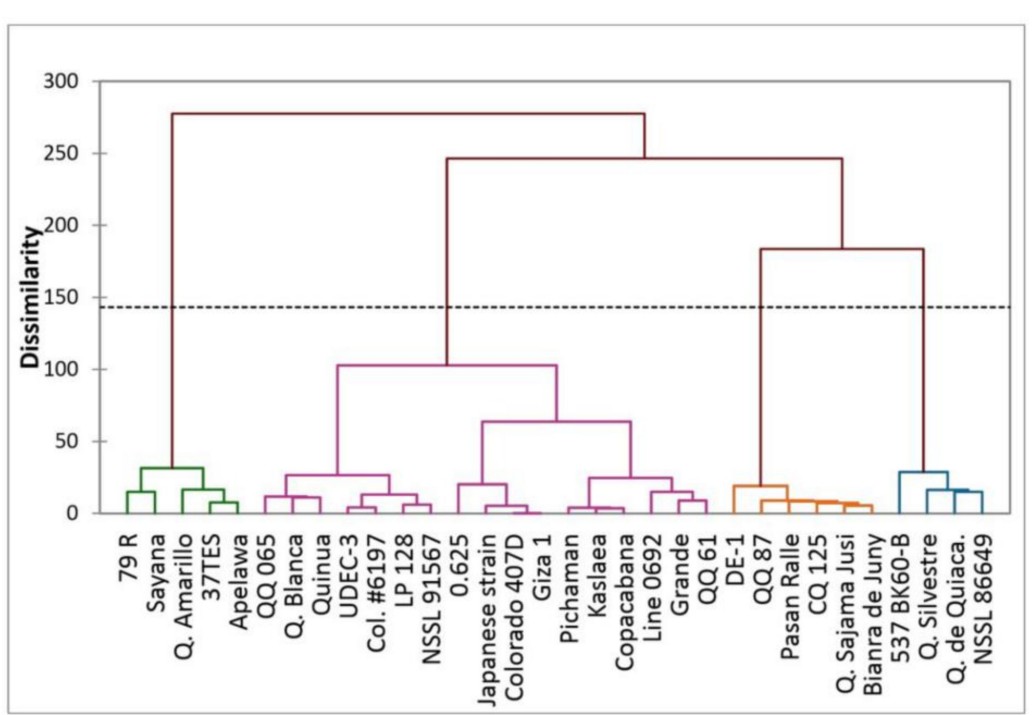

**Figure 3.** Agglomerative hierarchical clustering (AHC) of quinoa genotypes based on the morphological traits using the standardized Euclidean distances coefficient.

The genetic relationships among the 32 quinoa genotypes were assessed using data generated from SRAP markers. Based on Jaccard similarity coefficients, the pairwise correlations were used to generate the dendrogram using (UPGMA) Unweighted Pair-Group Method with Arithmetic means due to higher cophenetic correlation coefficient (Figure 4).

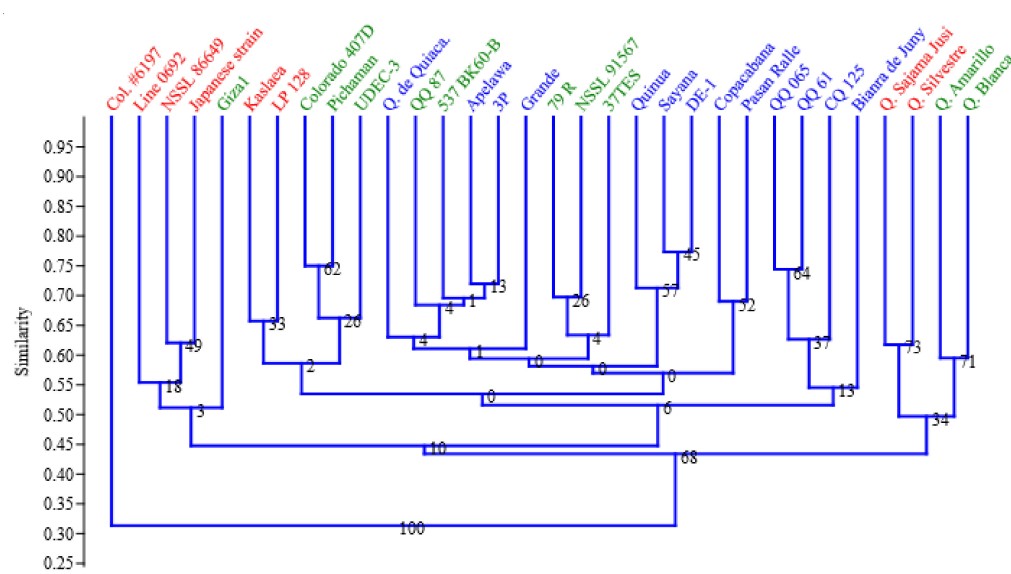

**Figure 4.** Dendrogram of the 32 quinoa genotypes based on SRAP markers using Jaccard's coefficient and the UPGMA clustering method. Numbers at the branches indicate bootstrap values, computed from 100 replications of the data.

**Table 4.** The features of sequence-related amplified polymorphism (SRAP) primers were selected regarding quinoa genetic diversity.

| Primer Combination | | Amplicons [a] | Total no. of Amplicon [b] | Average No. of Amplicon [c] | PIC Value | DP |
|---|---|---|---|---|---|---|
| Forward | Reverse | | | | | |
| ME19 | EM19 | 6 | 37 | 1.16 | 0.68 | 5.71 |
| ME21 | EM08 | 3 | 44 | 1.38 | 0.54 | 2.86 |
| ME21 | EM21 | 4 | 61 | 1.91 | 0.60 | 3.81 |
| ME29 | EM07 | 2 | 57 | 1.78 | 0.50 | 1.90 |
| ME29 | EM08 | 2 | 60 | 1.88 | 0.50 | 1.90 |
| ME29 | EM19 | 3 | 50 | 1.56 | 0.56 | 2.86 |
| ME30 | EM31 | 3 | 56 | 1.75 | 0.61 | 2.86 |
| ME05 | EM20 | 4 | 51 | 1.59 | 0.63 | 3.81 |
| ME05 | EM28 | 3 | 33 | 1.03 | 0.62 | 2.86 |
| ME05 | EM29 | 3 | 41 | 1.28 | 0.43 | 2.86 |
| ME05 | EM31 | 3 | 49 | 1.53 | 0.56 | 2.86 |
| ME09 | EM19 | 3 | 74 | 2.31 | 0.65 | 2.86 |
| ME15 | EM31 | 3 | 41 | 1.28 | 0.61 | 2.86 |
| ME20 | EM22 | 6 | 45 | 1.41 | 0.66 | 5.71 |
| ME20 | EM08 | 3 | 38 | 1.19 | 0.34 | 2.86 |
| ME27 | EM19 | 4 | 66 | 2.06 | 0.62 | 3.81 |
| ME27 | EM29 | 3 | 48 | 1.50 | 0.60 | 2.86 |
| ME27 | EM31 | 3 | 40 | 1.25 | 0.56 | 2.86 |
| ME26 | EM20 | 4 | 74 | 2.31 | 0.73 | 3.81 |
| ME27 | EM28 | 6 | 63 | 1.97 | 0.66 | 5.71 |
| ME28 | EM29 | 4 | 77 | 2.41 | 0.72 | 3.81 |
| Total | | 75 | 1105 | — | — | — |
| Mean | | 3.57 | 52.62 | 1.64 | 0.59 | 3.40 |
| Min | | 2.00 | 33.00 | 1.03 | 0.34 | 1.90 |
| Max | | 6.00 | 77.00 | 2.41 | 0.73 | 5.71 |

[a]: Total number of differently sized SRAP fragments amplified across all 32 genotypes; [b]: total number of SRAP fragments scored for all genotypes; [c]: average number of SRAP fragments scored per genotype. PIC: polymorphism information content. DP: discrimination power.

In this study, UPGMA cluster analysis exhibited weak clustering relationships (weak bootstrapping), which illustrated that genotypes tend to group in some cases according to the region of genotypes. The clustering patterns showed three distinct major clusters at 50% similarity, while only one genotype (Col. #6197, an improved genotype from the USA) separated and failed to form a cluster. This could be because of its origin, as it has crossed

or had variations induced by mutations and selection for several years. Both clusters I and II were composed of four genotypes, with the first cluster comprising four Peruvian cultivars (Q. Amarillo, Q. Sajama Jusi, Q. Silvestre, and Q. Blanca). In the second cluster, cultivars or improved genotypes from Egypt (Giza 1), Bolivia (Line 0692), and the USA (Japanese strain and NSSL 86649) were grouped. Cluster III comprised 23 genotypes and included the highest similarity (78%) between the genotypes of Sayana (Bolivian cultivar) and DE-1 (Ecuadorian cultivar). Fuentes et al. reported that ecological constraints had increased quinoa diversity and that cluster analysis discriminated between the central Andes genotypes and southern latitudes genotypes. Three European genotypes were grouped with the southern quinoa group [8].

Table 5 presents the analyzed genetic diversity parameters. The average number of alleles (Na) varied between 1.185 for population 2 to 1.95 for population 1. Population 1 presented greater mean values for the Shannon index (0.411), gene diversity (0.271), and polymorphic percentage (83.95). These genotypes presented more private alleles (10) compared with genotypes from population two and admixture. The analysis of molecular variance (AMOVA) showed that most of the genetic variation was according to differences within populations 86%). In comparison, the variability among populations was 14%, and the population differentiation was significant (PhipT = 0.14, $p < 0.001$).

**Table 5.** Classification of Quinoa genotypes and their inferred subpopulation identities.

| No. | Accessions | Inferred Clusters | Origin | Population ID |
|-----|-----------|-------------------|--------|---------------|
| 1 | Giza1 | admixture | Egypt | 1 |
| 2 | LP-128 | 2 | Argentina | 2 |
| 3 | Q-Silvestre | 2 | Argentina | 2 |
| 4 | Bianra-de-Juny | 1 | Australia | 3 |
| 5 | Apelawa | 1 | Bolivia | 4 |
| 6 | CQ-125 | 1 | Bolivia | 4 |
| 7 | Pasan-Ralle | 1 | Bolivia | 4 |
| 8 | Sayana | 1 | Bolivia | 4 |
| 9 | Line-0692 | 2 | Bolivia | 4 |
| 10 | Q-Sajama-Jusi | 2 | Chile | 5 |
| 11 | Pichaman | admixture | Chile | 5 |
| 12 | QQ-87 | admixture | Chile | 5 |
| 13 | UDEC-3 | admixture | Chile | 5 |
| 14 | DE-1 | 1 | Ecuador | 6 |
| 15 | Grande | 1 | Peru | 7 |
| 16 | Q-de-Quiaca | 1 | Peru | 7 |
| 17 | QQ-065 | 1 | Peru | 7 |
| 18 | QQ-61 | 1 | Peru | 7 |
| 19 | Quinua | 1 | Peru | 7 |
| 20 | Q-Amarillo | admixture | Peru | 7 |
| 21 | Q-Blanca | admixture | Peru | 7 |
| 22 | Col-#6197 | 2 | USA, Colorado | 8 |
| 23 | Colorado-407D | admixture | USA, Colorado | 8 |
| 24 | 537-BK60-B | admixture | USA, Maryland | 9 |
| 25 | 3P | 1 | USA, New Mexico | 10 |
| 26 | Copacabana | 1 | USA, New Mexico | 10 |
| 27 | Kaslaea | 2 | USA, New Mexico | 10 |
| 28 | 37TES | admixture | USA, New Mexico | 10 |
| 29 | 79R | admixture | USA, New Mexico | 10 |
| 30 | NSSL-91567 | admixture | USA, New York | 11 |
| 31 | NSSL-86649 | 2 | USA, South | 12 |
| 32 | Japanese-strain | 2 | USA, Washington | 13 |

The population structure of the 32 quinoa genotypes was inferred using STRUCTURE 2.3.4 [23], and the peak of delta K was observed at K = 2, suggesting the presence of two main populations (Figure 5A). The classification of genotypes into populations based on

the model-based structure from STRUCTURE 2.3.4 is shown in Figure 5B. The 32 genotypes were distributed to the main clusters and admixture. The first cluster of 13 (41%) of total genotypes was grouped into cluster one, the next 8 (25%) into cluster two, and 11 (34%) were placed in the admixture (Table 5).

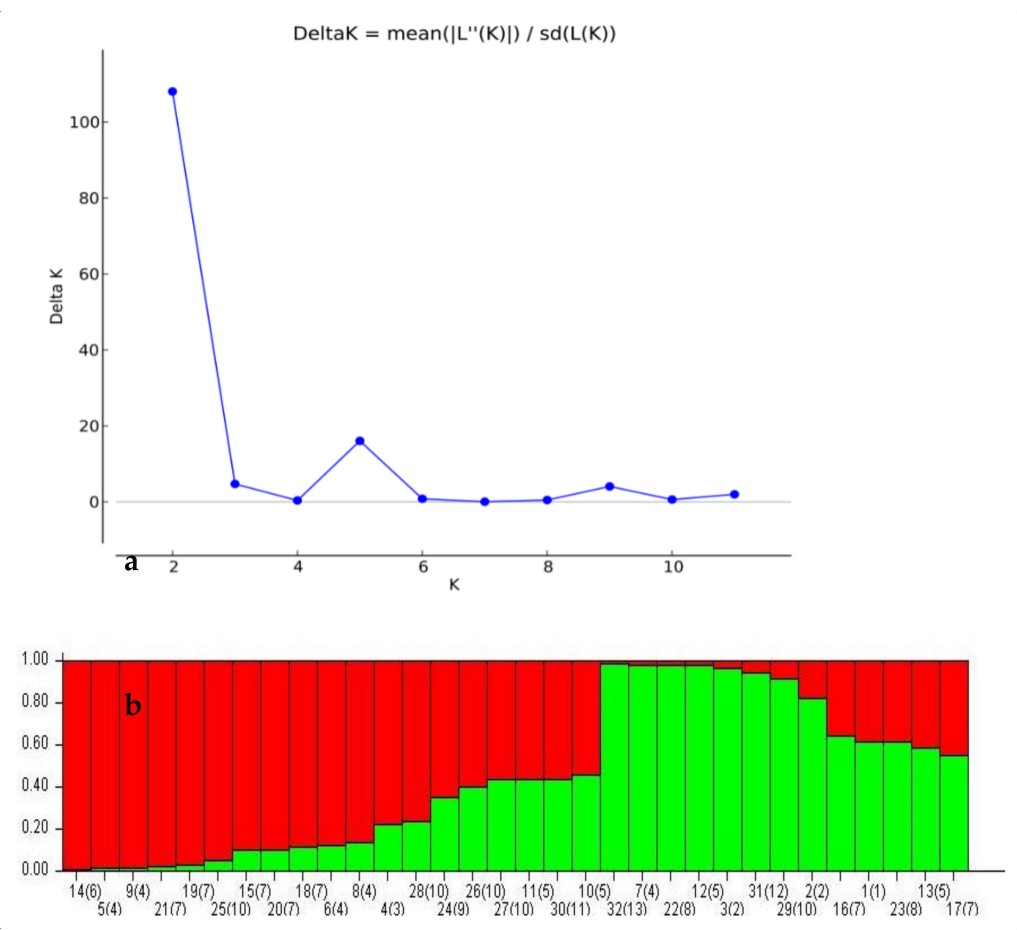

**Figure 5.** (**a**) ΔK values for each number of subpopulations (K) for 32 quinoa genotypes. (**b**) Classification of 32 quinoa genotypes into two main populations using STRUCTURE 2.3.4 software. The color code indicates the distribution of the genotypes to different populations. Numbers on the *y*-axis show the subgroup membership, and the *x*-axis shows the different genotypes and population ID in brackets.

We also performed principal coordinate analysis (PcoA) on 32 genotypes (Figure 6). This analysis largely supported the separation of the genotypes into two subpopulations fairly well distributed on the axes. The first three coordinates respectively contributed by 17.4, 13.0, and 10.2 of the total variation. Coordinate one clearly separated genotype of the population one from genotypes of the population two. The separation results were also evidenced in the model-based genetic clustering using STRUCTURE.

Using 15 species-specific SSR markers, the authors of [10] detected high genetic heterozygosity (0.71) for 84 accessions in the Ecuadorian Andes, showing that Ecuadorian quinoa is highly diverse. However, the authors of [11] reported He values ranging from 0.20 to 0.90, with a mean value of 0.57, in a panel of diverse quinoa accessions (31 cultivated quinoa accessions) representing the major quinoa growing areas of South America. Christensen et al. analyzed the genetic diversity of 152 quinoa accessions using 36 microsatellite loci and detected higher heterozygosity (He) values for the microsatellite loci with a range of 0.45–0.94 and a mean value of 0.75 [12]. Fuentes et al. reported using 20 SSR primers to analyze genetic diversity among 59 quinoa accessions from highland and coastal zones of Chile, with the coastal genotypes showing a higher quantity of alleles with a higher

value for Shannon index than highland genotypes [8]. Castillo et al. used RAPD to test the genetic diversity among Bolivian wild and cultivated quinoa varieties. The genetic diversity index (He) between the two quinoa forms was 0.288 for the weedy form and 0.311 for the cultivated form.

Regarding ecoregions, there were no apparent differences in genetic diversity either. They concluded that RAPD markers showed quinoa has a stable population structure and high intra-population variation [33]. Moreover, the authors of [34] reported values of 1.99, 0.50, and 0.69 for the number of alleles (Ne), Nei's genetic diversity (h), and Shannon's information index, respectively, for a French Vanilla quinoa genotype using inter-primer binding site markers.

The reason the factors behind the increasing number of alleles and increased gene diversity varied among genetic diversity studies could be because of the different molecular markers used (i.e., dominant vs. codominant markers), resolving the marker alleles using different electrophoresis systems in the fragmentation of amplified products, and the size and geographic composition of the tested germplasm.

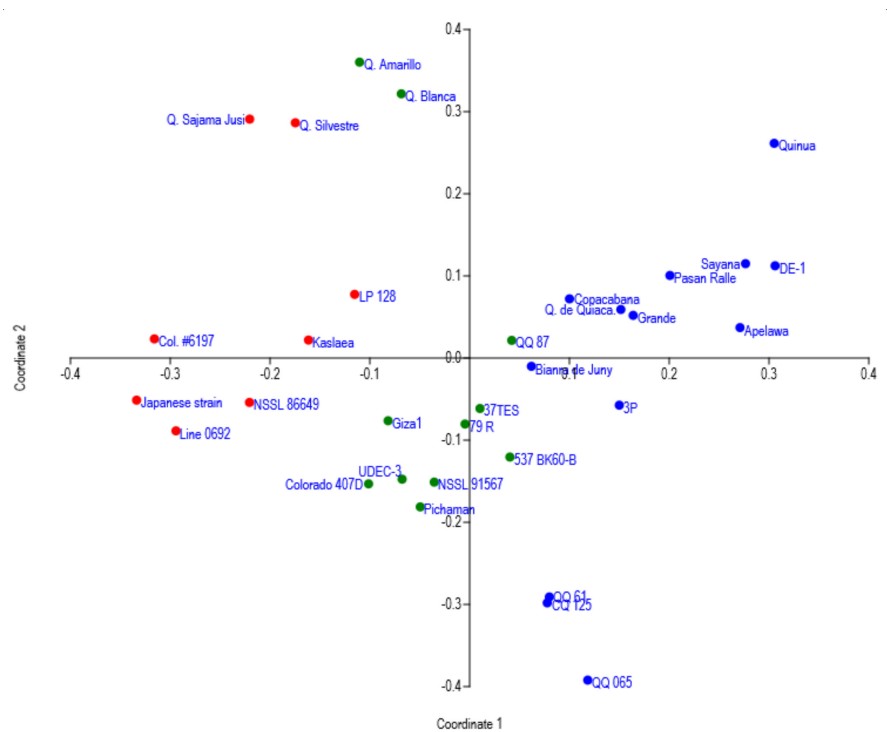

**Figure 6.** Principal coordinate analysis results confirming the presence of two main groups and one admixture. The first coordinate contributed by 17.4 and the second coordinate by 13.0 of the total variance.

Genetic differentiation has been observed (PhiPT = 0.14, *p* < 0.001) regarding the three populations (Table 6). Different studies have also been shown based on RAPD markers. A low level of intraspecific variation within *C. quinoa* and other Chenopodium species' accessions was reported. However, high levels of polymorphism were observed among all the species studied [35]. Furthermore, in [33], pure populations of cultivated and weedy quinoa were shown to have low differentiation levels. However, the differentiation among populations is much more critical than the differentiation between cultivated and weedy genotypes, which could be explained by the significant gene flows between weedy and cultivated quinoa measured at the field level.

**Table 6.** Genetic diversity parameters, analysis of molecular variance (AMOVA), and population differentiation (PhiPT) result for quinoa populations obtained via SRAP alleles analysis.

| Population | No. of Genotypes | Na | Ne | I | H | % P | Private Alleles |
|---|---|---|---|---|---|---|---|
| Population 1 | 13 | 1.765 | 1.457 | 0.411 | 0.271 | 83.95 | 10 |
| Population 2 | 8 | 1.185 | 1.324 | 0.283 | 0.188 | 56.79 | 2 |
| population 3 | 11 | 1.531 | 1.401 | 0.351 | 0.232 | 71.6 | 2 |
| Source | Df | SS | MS | Est. Var. | % | | |
| Among Pops | 2 | 57.047 | 28.523 | 1.717 | 14% | | |
| Within Pops | 29 | 305.984 | 10.551 | 10.551 | 86% | | |
| Total | 31 | 363.031 | | 12.268 | 100% | | |
| PhiPT | 0.14 | $p < 0.001$ | | | | | |

Note: Na = No. of alleles; Ne = No. of effective alleles; I = Shannon indexs; H = diversity; No. private alleles = No. of alleles unique to a single population; PhiPT = AP/TOT; AP = Est. Var. Among Pops.

The relationship between morphological and molecular distances measured by the Mantel test was non-significant (r = 0.01; $p > 0.58$) and was detected in quinoa by other markers. Rodríguez and Isla found no correlations between morphological and molecular marker results using AFLP markers [36]. However, they discovered the possibility of finding molecular differences or similarities correlated with some morphological traits, i.e., grain color and panicle color. The PIC values measured for all SRAP primers were high and ranged from 0.34 to 0.73, with a mean value of 0.59, while primers showed discrimination powers (DP) ranging from 1.9% to 5.7%. The PIC values were classified as high informative (PIC > 0.5), intermediate informative (0.25 < PIC < 0.5), and low informative (PIC < 0.25) [19]. Primer combinations 4 and 5 had the lowest discrimination powers, while primer combinations ME19/EM19, ME20/EM22, and ME27/EM28 performed the best. Few studies have developed and use molecular markers in quinoa, and there has been a particular lack of such studies on the lowland quinoa genotypes [8]. To the best of our knowledge, we report for the first time using SRAP markers to characterize quinoa genotypes, including different ecotypes. A high level of diversity was observed, which agrees with the results recorded previously [8]. An average PIC value of 0.28 was recorded in ornamental pomegranates [37]. The phenotypic or molecular markers were used to investigate quinoa's population structure and diversity in several previous studies [8,9,12]. Bhargava et al. mentioned that hybridized quinoa genotypes from different clusters with different desirable genes for a specific trait could be facilitated to accumulate favorable genes of the trait in new lines [28]. However, it is crucial to consider inter-cluster distances. The magnitude of hybrid vigor depends on the diversity between the two clusters' parental and statistical distances. Using genome re-sequencing data from 21 *Chenopodium* accessions, [6] investigated single-nucleotide polymorphisms (SNPs) and copy number variation (CNV); they identified 15 genes that could contribute to the differences in salinity tolerance of these quinoa accessions.

## 4. Conclusions

This study showed significant genetic variability between quinoa genotypes. This considerable phenotypic variation observed among the 32 accessions is remarkable and means this group of accessions is of great importance and might represent a basis for the quinoa breeding program to select early and adapted genotypes. Genetic diversity parameters and differentiation revealed significant differences and could be used in future genetic research. Morphological and molecular markers confirm such differences. The clustering pattern based on molecular markers was not congruent with that of morphological-based markers. Quinoa genotypes are grouped according to their origin or genetic background via both molecular and phenotypic assessments. These clustering patterns will be used in future research, including conservation, molecular assist breeding, and phytochemical and physiological studies.

**Supplementary Materials:** The following are available online at https://www.mdpi.com/article/10.3390/agriculture11040286/s1, Table S1: Name and sequence of selected SRAP primer used, Table S2: Description of the thirty-two quinoa genotypes through seventeen qualitative traits, Table S3: Mean of the eleven quantitative traits with the standard errors for the evaluated thirty-two quinoa genotypes.

**Author Contributions:** Conceptualization, E.H.E.-H. methodology, E.H.E.-H.; A.G.; T.K.A.; M.A. software, H.M.M.; M.A.K. validation, E.H.E.-H.; S.S.A. and H.M.M.; formal analysis, E.H.E.-H.; H.M.M. investigation, S.S.A. data curation, M.A.K.; M.A.; S.A.A.-F. writing—original draft preparation, E.H.E.-H.; H.M.M. writing—review and editing, H.M.M. All authors have read and agreed to the published version of the manuscript.

**Funding:** Deanship of Scientific Research, King Saud University grant number RG-1441-513.

**Institutional Review Board Statement:** Not applicable.

**Informed Consent Statement:** Not applicable.

**Data Availability Statement:** Not applicable.

**Acknowledgments:** The authors extend their appreciation to the Deanship of Scientific Research at King Saud University for funding this work through research group no. RG-1441–513.

**Conflicts of Interest:** The authors declare no conflict of interest.

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
