# Peer review of "Morphological and Molecular Characterization of Quinoa Genotypes"

_agriculture, doi:10.3390/agriculture11040286_

Round 1

Reviewer 1 Report

The paper now is of higher quality, but still I would suggest slightly change the layout of Table 2 - it is hardly understandable due to some unnecessary `enters` used between lines or too narrow columns. The table looks careless.

Author Response

The paper now is of higher quality, but still, I would suggest slightly change the layout of Table 2 - it is hardly understandable due to some unnecessary `enters` used between lines or too narrow columns. The table looks careless.

Thank you very much, we have improved the layout of the table according to your suggestions

Reviewer 2 Report

My main points of concern for manuscript by El-Harty et al are:

1. Why authors use Euclidean distance for hierarchical clustering? Did they try any other distance metric? For example Manthatan distance?

2. On page 9 the authors wrote "Morphological traits are used to assess the genetic diversity between quinoa genotypes grouped using Euclidean distance". If the morphological traits were used shouldn't it be morphological diversity instead of genetic diversity?

3. The authors do not provide any information about numbers of branches on figure 4. My guess is that they represent bootstrap values. If they are bootstrap values for some branches they are very low like 0, 1 or 7. What is the point of describing so unstable tree?

4. On page 12 authors wrote "The genetic relationships among the 32 quinoa genotypes were assessed using a neighbor-joining algorithm with the unweighted pair group method (UPGMA) using SRAP markers (Figure 4)." How it is possible if those are two separate algorithms for constructing genetic tree?

5. In the AMOVA analysis proposed population structure (divided into two groups) explained about 4% of total observed variability. Why authors think this structure is any good? Why they don't try finding any better genetic structure using statistical methods for example k-means?

6. Are the AMOVA results statistically significant?

Author Response

Dear Respective professor

Thank you very much for the comment raised that improved our manuscript more and more. Please find attached our response for your kind attention.

Round 2

Reviewer 2 Report

I am pleased to see the amendments made by the authors. However, in my opinion, the authors should make a division of the studied gentypes according to the new structure of the population. The use of Structure software as proposed by the authors is one of the possible solutions. Once the new structure is established, it should be assessed using the AMOVA technique. In my opinion, such results will explain significantly more of the observed variability than the currently reported 4%. As a result, the value of the article will be increased.
Additionally, in my opinion, a comment about the bootstrap value should be included in the text of the manuscript. Authors should comment on the results obtained in bootstrap analysis in the discussion section.

Author Response

Dear Respective Proposer

Thank you very much for the valuable comments you suggested. Reanalysing data using STRUCTURE software added valuable results and explained significantly more of the observed variability present in the genotypes. Moreover, AMOVA was reanalyzed and the results became more reliable. All changes are highlighted in different colors in the manuscript. Again thank you very much.

Best regards

This manuscript is a resubmission of an earlier submission. The following is a list of the peer review reports and author responses from that submission.

Round 1

Reviewer 1 Report

The research topic is of high interest for the breeding programs and covers both aspects of evaluation of genetic diversity - morphological and molecular. Appropriate methodological approach is used to reach the aim of the paper - to characterize 32 introduced genotypes of quinoa grown under the semiarid climatic conditions of Saudi Arabia.

Nonetheless several remarks/suggestions to authors for paper improvement are given to consideration:

1) the style of typing the references in the text in some places is imprecise, e.g. [11; 12; [8]; 9].

2) the text would be easier perceivable if quoted authors would be mentioned by surnames and then follows reference number in brackets at the end of sentence. For example, sentence  "[9] reported that breeding to improve quinoa was restricted because of a lack of genetics and genomics information about the crop." seems strange. In stead I would suggest to use "Zhang and colleagues reported that breeding to improve quinoa was restricted because of a lack of genetics and genomics information about the crop [9]."

3) There is incorrect chemical formula for Calcium superphosphate provided in the Materials and Methods description (P2O5).

4) The information about watering once per month does not reflect anything - it would be good to give at least water amount or soil moisture level to get idea does it is enough to water plants once  a month in SA conditions.

5) Wording and typing would be good to check by English editor.

Author Response

  1. The style of typing the references in the text in some places is imprecise, e.g. [11; 12; [8]; 9].

Thank you, all references cited in the text are corrected

  1. The text would be easier perceivable if quoted authors would be mentioned by surnames and then follows reference number in brackets at the end of sentence. For example, sentence  "[9] reported that breeding to improve quinoa was restricted because of a lack of genetics and genomics information about the crop." seems strange. In stead I would suggest to use "Zhang and colleagues reported that breeding to improve quinoa was restricted because of a lack of genetics and genomics information about the crop [9]."

Thank you, all are corrected.

  1. There is incorrect chemical formula for Calcium superphosphate provided in the Materials and Methods description (P2O5).

            Thank you, the formula is corrected,

  1. The information about watering once per month does not reflect anything - it would be good to give at least water amount or soil moisture level to get idea does it is enough to water plants once  a month in SA conditions.

Thank you, plants were irrigated according to their performance and not to subjected to stress. So, plants were watered once a week rather than once a month

  1. Wording and typing would be good to check by English editor.

Thank you, the manuscript was checked, and language edited

Reviewer 2 Report

Results are interesting, but there is a lot of issues in writing - references, tables and some figures seem unfinished (the ones from PAST need refinig, they are not 300 DPI), results and discussion sections are not separate so it needs to be re-written, English needs improvement, etc.

Author Response

  1. There is a lot of issues in writing - references,

Thank you for all references were edited according to journal instruction

  1. Tables and some figures seem unfinished (the ones from PAST need refining, they are not 300 DPI)

Thank you, the figures and tables were improved  

  1. Results and discussion sections are not separate so it needs to be re-written,

Thank you, according to Jornal instruction, the discussion section may be combined with Results

  1. English needs improvement, etc

Thank you, the manuscript was checked, and language edited 

Reviewer 3 Report

MS: AGRICULTURE-1100339

 MORPHOLOGICAL AND MOLECULAR CHARACTERIZATION OF QUINOA GENOTYPES

 The paper report on performance of 32 quinoa genotypes, tested using morphological and molecular traits under Saudi Arabia conditions and provides the basic information necessary for breeding of this species.

Genotypes were compared based on morphological and qualitative and quantitative characteristics using cluster analysis and also by means of SRAP molecular markers.

The manuscript is generally well written and clearly presented.

I believe this paper could have good value for the researchers engaged in the agrobiodiversity sector worldwide and in the quinoa genetic improvement sector in particular.

There are some areas in need of improvement in the manuscript, and they are detailed here and in the attached pdf file.

For this, itshould have some following MINOR REVISIONS below stated.

I suggested the Authors make some revisions (comments) present as note and yellow highlights in the attached modified pdf file, and here listed.

  1. Make sure that the abstract is informative, can stand alone and covers the content. Please improve it, as in this form, according to me, it does not meet all the needs exposed.
  2. Abstract: Please better explain the yellow highlighted sentence.
  3. Introduction (Pag.1): Please improve the yellow highlighted sentence: in this form reading is difficult. According to me after every ecotype name ":" are better than ";" i.e. ”Valley: grown at 2000 to 3500 m; altiplano: grown at high altitudes of over 3500 km, around Titicaca Lake;” and so...  
  4. Introduction (Pag.1): “South America” and other similar countries all over the world, I suppose... Please improve the yellow highlighted sentence.
  5. Introduction (Pag.1): “[6]” According to me the name of the Author(s) should be cited in the text and the reference apposed as is [6], f.i. This tip is not based on grammar error, but rather the conventions of academic scientific writing...  Please check other similar tips: “[9]” and others.
  6. Introduction (Pag.1): “These” - Which? Please better explain what does the word "these" refer to.
  7. In the attached modified pdf file please check all the yellow highlighted sentence (or words) and change or confirm.

Author Response

  1. Make sure that the abstract is informative, can stand alone and covers the content. Please improve it, as in this form, according to me, it does not meet all the needs exposed.Abstract: Please better explain the yellow highlighted sentence.

Thank you, the abstract was improved, and all comments are considered

  1. Introduction (Pag.1): Please improve the yellow highlighted sentence: in this form reading is difficult. According to me after every ecotype name ":" are better than ";" i.e. ”Valley: grown at 2000 to 3500 m; altiplano: grown at high altitudes of over 3500 km, around Titicaca Lake;” and so

Thank you, the sentence is rephrased and improved

  1. Introduction (Pag.1): “South America” and other similar countries all over the world, I suppose... Please improve the yellow highlighted sentence.

Thank you, we considered your comment

  1. Introduction (Pag.1): “[6]” According to me the name of the Author(s) should be cited in the text and the reference apposed as is [6], f.i. This tip is not based on grammar error, but rather the conventions of academic scientific writing...  Please check other similar tips: “[9]” and others.

Thank you, all were corrected.

  1. Introduction (Pag.1): “These” - Which? Please better explain what does the word "these" refer to.

Thank you, it was clarified

  1. In the attached modified pdf file please check all the yellow highlighted sentence (or words) and change or confirm.

Thank you, all comments are considered and were clarified and corrected

Reviewer 4 Report

The manuscript article by El-Harty et al. entitled "Morphological and Molecular Characterization of Quinoa Genotypes" is trying a good attempt to research scientifically, but there are some following serious flaws :

The study is very superficial and basic and such studies have already been done by other researchers worldwide.  This tools and techniques are not novel to incorporate in the scientific manuscript. The overall writing is not good. For population genetic studies, 32 genotypes are very little and insufficient to formulate any population structure. Primer sequence table is also unnecessary in the main manuscript. Also figure 1 doesn’t have any bar for each picture. There is no scale in figure 3 dendrogram, even such clustering based on the morphometric traits are so obsolete in this century.  

Author Response

  1. The study is very superficial and basic and such studies have already been done by other researchers worldwide. This tools and techniques are not novel to incorporate in the scientific manuscript.

Thank you, quinoa is introduced to Saudi Arabia for the first time, and we have to evaluate these genotypes under Saudi environments.  After evaluation, selected genotypes will go through a breeding program for improving quantity and quality parameters. Fore that genetic variability studies at the morphological and molecular levels have to be conducted.

  1. The overall writing is not good.

Thank you, the authors were going thoroughly the manuscript and tried to improve the quality of the manuscript.

  1. Primer sequence table is also unnecessary in the main manuscript.

Thank you, the table is moved to the supplementary tables

  1. Also figure 1 doesn’t have any bar for each picture.

Thank you, the figure is improved

  1. There is no scale in figure 3 dendrogram, even such clustering based on the morphometric traits is obsolete in this century.

Thank you, the scale is according to Eucledain distance values.

Round 2

Reviewer 2 Report

Comments are in the attached file. Please use the file only to see the comments. I transformed the pdf to docx so this file should be only used as a guideline and not to be submitted as final version after corrections.

Author Response

Thank you very much for your efforts to improve our manuscript. Please find attached the file regarding the response to each comment.

Reviewer 4 Report

Still, the manuscript is not improved scientifically as per my previous comment.

Author Response

  1. The study is very superficial and basic and such studies have already been done by other researchers worldwide. ​​​​​​​These tools and techniques are not novel to incorporate in the scientific manuscript.

 Thank you, we have introduced quinoa genotypes to Saudi Arabia to study the performance and stability of growing quinoa in dried environments. Because our area lacking genotypes and our farmers are never practicing the cultivation of this crop, we must evaluate and demonstrate the cultivation of this crop in our area to be a new food and feed crop, and farmers in dry areas can cultivate this crop in rainfed and irrigated environments.

Moreover, the Saudi Ministry of water, environment, and agriculture developed initiatives projects to encourage farmers to cultivate this crop. For that, we directed our research to do evaluation and next to involve quinoa in breeding programs.

  1. The overall writing is not good.

Thank you, the authors were going thoroughly the manuscript and tried to improve the manuscript's quality.

  1. The primer sequence table is also unnecessary in the main manuscript.

Thank you, the table is moved to the supplementary tables

  1. Also figure 1 doesn't have any bar for each picture.

Thank you, the figure is improved, and the scale was added to show the height in cm.

  1. There is no scale in figure 3 dendrogram, even such clustering based on the morphometric traits is obsolete in this century.

Thank you, the scale is according to Eucledain distance values.